# Genome-Wide Analysis of *Nuclear factor-YC* Genes in the Tea Plant (*Camellia sinensis*) and Functional Identification of *CsNF-YC6*

**DOI:** 10.3390/ijms25020836

**Published:** 2024-01-09

**Authors:** Shengxiang Chen, Xujiao Wei, Xiaoli Hu, Peng Zhang, Kailin Chang, Dongyang Zhang, Wei Chen, Dandan Tang, Qian Tang, Pinwu Li, Liqiang Tan

**Affiliations:** 1College of Horticulture, Sichuan Agricultural University, Chengdu 611130, China; csx810905@163.com (S.C.);; 2Tea Refining and Innovation Key Laboratory of Sichuan Province, Chengdu 611130, China

**Keywords:** *Camellia sinensis*, *NF-YC*, expression analysis, growth and development, abiotic stress

## Abstract

Nuclear factor Y (NF-Y) is a class of transcription factors consisting of NF-YA, NF-YB and NF-YC subunits, which are widely distributed in eukaryotes. The NF-YC subunit regulates plant growth and development and plays an important role in the response to stresses. However, there are few reports on this gene subfamily in tea plants. In this study, nine *CsNF-YC* genes were identified in the genome of ‘Longjing 43’. Their phylogeny, gene structure, promoter cis-acting elements, motifs and chromosomal localization of these gene were analyzed. Tissue expression characterization revealed that most of the *CsNF-YCs* were expressed at low levels in the terminal buds and at relatively high levels in the flowers and roots. *CsNF-YC* genes responded significantly to gibberellic acid (GA) and abscisic acid (ABA) treatments. We further focused on *CsNF-YC6* because it may be involved in the growth and development of tea plants and the regulation of response to abiotic stresses. The CsNF-YC6 protein is localized in the nucleus. Arabidopsis that overexpressed *CsNF-YC6* (*CsNF-YC6-OE*) showed increased seed germination and increased root length under ABA and GA treatments. In addition, the number of cauline leaves, stem lengths and silique numbers were significantly higher in overexpressing Arabidopsis lines than wild type under long-day growth conditions, and *CsNF-YC6* promoted primary root growth and increased flowering in Arabidopsis. qPCR analysis showed that in *CsNF-YC6-OE* lines, flowering pathway-related genes were transcribed at higher levels than wild type. The investigation of the *CsNF-YC* gene has unveiled that *CsNF-YC6* plays a pivotal role in plant growth, root and flower development, as well as responses to abiotic stress.

## 1. Introduction

Under the influence of variable environments, plants have evolved a complex set of regulatory mechanisms in which the nuclear factor Y (NF-Y) plays an important role. NF-Y is a conserved class of heterotrimers found in eukaryotes and distributed in all known sequenced eukaryotes, consisting of NF-YA, NF-YB and NF-YC [1]. The NF-Y complex regulates the expression of target genes by discriminating and binding to their CCAAT sequences in the promoter region to regulate their transcription [2]. Unlike mammals and fungi where a single subunit is encoded by only a single gene, NF-Y subunits in plants are usually encoded by multiple genes [3]. In Arabidopsis (*Arabidopsis thaliana*), 10 *NF-YA*, 13 *NF-YB* and 13 *NF-YC* were identified [4]; *NF-YA*, *NF-YB* and *NF-YC* in rice (*Oryza sativa*) were 10, 11 and 7, respectively [5]; 10, 22 and 9 in potato (*Solanum tuberosum* L.) [6]; 10, 11 and 14 in wheat (*Triticum aestivum*), respectively [7] and 6 *NF-YA*, 12 *NF-YB* and 6 *NF-YC* were identified in peach (*Prunus persica* L.) [8].

The different subunits of NF-Y transcription factors all contain a relatively conserved structural domain capable of binding DNA or interacting with the protein. Among them, the NF-YC protein is located in the middle of the three proteins in size, and its conserved domain also contains an histone fold motif (HFM) structural domain, similar to that of the H2A protein. This conserved domain mainly includes two β-strands and three α-helices (α1, α2, α3) within the HFM structural domain and one α-helix (αC) outside the HFM folding region, which are 7, 27, 8 and 7 amino acids in length, respectively, and are involved in protein–protein or protein–DNA interactions [9].

NF-Ys regulates the entire growth and development process of plants, from vegetative growth to reproductive growth, including seed germination, root growth and plant flowering [10,11,12,13,14,15,16]. In addition, *NF-Ys* are involved in the regulation of stress response to low temperature, drought, salinity and high heat. *AtNF-YC2* was the first plant *NF-YC* member to be isolated and was shown to play an important role in the flowering process in Arabidopsis [17]. The Arabidopsis *NF-YC* was found to form the NF-YC-YB-CO/YA complex, which recognizes the CCAAT-box enhancer distal to the flowering locust T (FT) promoter and the proximal co-response element (CORE) to increase the transcriptional level of FT and thus play a role in photoperiod-dependent flowering [18,19]. In rice, *NF-YC2*, *NF-YC4* and *NF-YC6* regulate photoperiodic flowering response and promote flowering by mediating the expression of GRAS proteins Ehd1, RFT1 and Hd3a [20]. *TaNF-YC5*, *TaNF-YC8*, *TaNF-YC9*, *TaNF-YC11* and *TaNF-YC12* in wheat and *AtNF-YC3*, *AtNF-YC4* and *AtNF-YC9* in Arabidopsis are regulated by light signals and involved in the CO-mediated photoperiodic and gibberellin pathways [21]. With the in-depth study of *NF-Y*, it was found that *NF-YC* can be involved in abiotic stress responses through complex signaling pathways, thus enhancing plant adaptation. For example, overexpression of *MsNF-YC2* enhanced alkali tolerance, antioxidant enzyme activity and proline content in transgenic alfalfa (*Medicago sativa* L.) plants [22]. *GmNF-YC14* and *Cdt-NF-YC1* can improve the drought and salt tolerance of transgenic soybean and rice through activating ABA-dependent and ABA-independent signaling pathways [10,23].

The tea plant (*Camellia sinensis* (L.) O. Kuntze) is widely grown worldwide as a perennial leaf-used evergreen economic plant. The flower development, winter dormancy and abiotic stresses affect the quality and yield of tea plants. Therefore, it would interesting to identify the *NF-YC* genes of the tea plants and character their roles in development and abiotic stresses. In this study, nine *CsNF-YC* genes were identified based on the reference genome; bioinformatics and expression analyses of *CsNF-YCs* were performed to explore their biological functions. In addition, *CsNF-YC6*, a member with an interesting expression profile, was detailed characterized via allogeneic overexpression in Arabidopsis. The results of this study provide clues to understand the roles of *CsNF-YCs* in the development and response to abiotic stresses of tea plants.

## 2. Results

### 2.1. Identification and Characterization of CsNF-YC Subunits in the Tea Plant Genome

Using the amino acid sequence of AtNF-YCs as references, a total of nine CsNF-YC family members were identified, and the CsNF-YCs were named CsNF-YC1 to CsNF-YC9. NF-YC family members were localized on nine different chromosomes, with gene sequence lengths ranging from 381 to 1443 bp. The length of the encoded proteins ranged from 126 to 480 aa, with theoretical isoelectric points and molecular masses ranging from 5.33 to 10.01 and 13.67 to 52.95 KDa (Appendix A).

The conserved structural domain analysis of CsNF-YCs has a distinct CBFD_NFYB_HMF conserved structural domain containing about 80 amino acids, similar to the histone H2A conformation domain, and a CCAAT binding region unique to the NF-YC family, which contains two NF-YA interaction domains and one NF-YB interaction domain and can form heterotrimers with NF-YA and NF-YB (Figure 1A).

Protein sequences of the *CsNF-YC* gene were used to construct a phylogenetic tree together with the NF-YCs from Arabidopsis, oilseed rape and Tribulus alfalfa (Figure 1B). The NF-YC members of tea plant and Arabidopsis crossed together without forming separate branches, indicating that the tea plant CsNF-YCs are more evolutionarily conserved and have similar structures and functions. CsNF-YC6 and CsNF-YC3 are more closely related to AtNF-YC3, AtNF-YC9 and MtNF-YC1.

All members of the CsNF-YC subfamily contain Motif 1 and Motif 3, which can be considered as core motifs with a structure similar to that of the H2A core histone folding motif; in addition, excluding CsNF-YC7 and CsNF-YC8, the other members all contain Motif 2 or Motif 4, and CsNF-YC1, CsNF-YC4, CsNF-YC3 and CsNF-YC6 all contain Motif 5 (Figure 1C). Among them, the CsNF-YC7 protein and CsNF-YC8 protein have essentially the same Motif type and arrangement, presumably with functional similarity. CsNF-YC1 contained the most motifs (9), while CsNF-YC7 and CsNF-YC8 contained the least motifs (3). Therefore, the type and number of conserved motifs were slightly different in distribution among different proteins, indicating that the degree of evolution of the tea plant NF-YC protein family is also somewhat different. The gene structure map shows that the number of exons of *CsNF-YC* genes generally contains two to three exons except *CsNF-YC6*, which had six exons (Appendix A).

A total of 65 different types of cis-elements were obtained in the promoter of tea-plant *CsNF-YC* genes, which were classified into three main categories, except for some cis-elements related to protein binding sites and the core of CAAT-box and TATA-box (Figure 1D). One category is stress response and cis-elements of phytohormone response, for example, there are seven low temperature responses (LTRs), five drought stress responses (MBSs) and fifteen ABA responses (ABREs). These include LTRs, MBSs, ABREs and other stress responses (STREs, TC-rich), GA responses (GARE-motif, P-box, and TATC-box), MeJA responses (CGTCA-motif, TGACG-motif) and ethylene responses (EREs). Another category is the large number of light response elements, such as 30 Box4 elements, 13 G-box elements, 12 GT1-motifs, 9 TCT-motifs, and 5 TCCC elements. There is also a category of elements related to plant growth and development, such as circadian, meristematic tissue expression (CAT-box), endosperm expression (GCN4) and damage repair (WUN-motif) elements. Among the total related components, the largest proportion was of stress-responsive components, and it is speculated that *CsNF-YCs* have a potential role in plant hormone and stress responses.

The sequences of eight *NF-YCs* were obtained by cloning from EW. Bioinformatics analysis of the eight *CsNF-YC* genes obtained from tea plant EW, such as the full length of the open reading frame, the number of encoded amino acids and the theoretical isoelectric point, was performed (Appendix A). The signal peptide prediction of NF-YC proteins showed that CsNF-YC was predicted to be in the nucleus. The secondary structure analysis of tea plant CsNF-YC protein was then performed, and it was found that the secondary structure of CsNF-YC protein consisted of α-helix, β-turn and irregular curl, mainly irregular curl and α-helix, and the proportion of irregular curl was the largest. Three-dimensional structural analysis showed that CsNF-YC contains a core conserved CBFD_NFYB_HMF structural domain containing four α helices (α1, α2, α3, αC) and two β chains.

### 2.2. Expression Patterns of CsNF-YC Genes in Different Tissues and Their Responses to Abiotic Stresses

Most of the *CsNF-YCs* were expressed at low levels in the terminal buds, while they had relatively high expression levels in the flowers and roots (Figure 2A–I). Among them, *CsNF-YC1*, *CsNF-YC4* and *CsNF-YC5* had less variation in expression in different tissues and lower expression variation. In comparison (using the flower buds as a control), *CsNF-YC2*, *CsNF-YC7* and *CsNF-YC8* had higher transcript levels in roots, with expression ploidy as high as 186.31, which may regulate the growth and development of the root system of tea plants. In addition, *CsNF-YC1*, *CsNF-YC3*, *CsNF-YC6* and *CsNF-YC8* had higher expression levels in floral tissues and may be involved in regulating floral meristematic tissues. the expression of *CsNF-YC8* in stems was 65-fold higher than that of the control, while all other *CsNF-YC* members showed lower expression levels.

Plant hormones play an important role as a signaling molecule in plant growth and development and in response to adversity. Under the induction of the hormone GA, the expression of each gene in the samples taken after 0 h of treatment was used as the control, except for *CsNF-YC9* and *CsNF-YC7*, all the *CsNF-YC* genes in the leaves showed up-regulated expression, and their expression patterns showed two cycles of “up-down” expression trends. Overall, most of the *CsNF-YCs* in tea leaves were up-regulated under GA induction (Figure 2L). Under the induction of the hormone ABA and all the *CsNF-YC* genes showed a decreasing and then increasing expression trend except for *CsNF-YC7* (Figure 2M). Probably due to the antagonistic effect of ABA and GA, the expression trend of *CsNF-YCs* was opposite to that induced by GA at the early stage of ABA induction (0–12 h of ABA treatment), and its expression was significantly down-regulated. And *CsNF-YC6* showed an increasing trend in its expression after 12 h of ABA treatment, which was earlier than the response time of the remaining *CsNF-YCs*. However, overall, the expression of most *CsNF-YCs* was higher than that under normal conditions after 24 h of GA and ABA hormone treatment.

Based on the expression studies of different tissues of tea plant EW, different hormone treatments and the transcriptome of the subject during the overwintering period [24], it was found that *CsNF-YC6* may be involved in the regulation of growth and development of tea plants and in response to abiotic stresses of adversity. Mature leaves and axillary bud tissues of tea plants were used to determine its expression levels in tea plants EW and CC2. In the leaves of tea plant EW, *CsNF-YC6* expression showed a trend of first increasing and then decreasing, and then increasing and then decreasing, and reached a maximum of 20.84-fold on February 3 (relative to the CC2 leaves on November 21). The expression of *CsNF-YC6* gene in tea plant CC2 leaves showed a trend of increasing and then decreasing, and there was no significant increasing trend in January and February (Figure 2J). In general, the relative expression of *CsNF-YC6* in tea plant EW leaves during overwintering was higher than that in tea plant CC2 leaves during the same period.

In the axillary buds of both tea plant EW and CC2, the expression of *CsNF-YC6* showed a trend of increasing and then decreasing, with a higher expression of 42.01 in the axillary buds of CC2 in December, while the axillary buds of tea plant EW remained at a relatively low level throughout the overwintering period, without significant changes (Figure 2K).

Overall, the relative expression of *CsNF-YC6* in CC2 leaves of tea plants increased significantly to reach the maximum value of 2.37-fold during the whole overwintering period on December 6, and the relative expression level of *CsNF-YC6* in CC2 axillary buds also increased significantly on December 20, indicating that the change in *CsNF-YC6* in the axillary buds of tea plants in response to the environmental temperature was later than that in tea leaves. Other studies have found that FT proteins can move from leaves to apical induced meristematic tissues to promote growth.

The transient expression assay of tobacco epidermal cells showed that the CsNF-YC6-eGFP fusion protein was expressed mainly in the nucleus (Figure 2N).

### 2.3. Morphological Characteristics of CsNF-YC6 Transgenic Arabidopsis Plants

To investigate the effect of *CsNF-YC6* gene expression on plant growth, we constructed Arabidopsis plants with *35S:CsNF-YC6-pCAMBIA1302* gene. Three T3 generation transgenic Arabidopsis lines (#1, #2 and #3) with high *CsNF-YC6* expression levels were randomly selected for analysis (Figure 3A). Under long daylight conditions (16 h/8 h), the overexpression plants twitched when the plants grew to 24 d (Figure 3B); when the plants grew to 30 d, we found that the overexpression plants twitched and flowered or started to grow horn fruits, while the wild-type plants were at the nutritional growth stage or twitching stage. At the same time, the stem length was also significantly higher than in the wild type (Figure 3C), indicating that *CsNF-YC6* promotes the growth and development of Arabidopsis.

To investigate the reasons and pathways for their early flowering, we selected healthy rosette leaves from the same parts of wild-type and overexpression lines and performed RNA extraction and fluorescence quantitative PCR to analyze the relative expression of the genes related to the flowering pathway (FLC, SOC1, CO, FD, FT, LFY and AP1) in these plants (Figure 3D). The results showed that in different transgenic lines, the relative transcript levels of flowering pathway-related integrator genes such as *FT* and *SCO1* and meristem-specific genes such as *LFY* and *AP1* were increased to different degrees, and the relative expression of *FLC* flowering repressor was significantly decreased (0.28, 0.41 and 0.47 in the three lines, respectively). The expression of *CO* and *FT* was significantly up-regulated in the expression of *FD* genes, which were shown to interact with *FT*, which, as indicated, was significantly up-regulated (4.06, 9.04 and 4.76 for the three lines, respectively). This indicates that *CsNF-YC6* can activate a series of flowering-related genes, i.e., the flowering promoter *CO* gene that integrates photoperiodic and biological clock signals, and induces the expression of *FT* and *SOC1* genes, thus inducing the expression of *FMI-like* genes (*LFY*, *AP1*), which in turn affect plant flowering.

### 2.4. Overexpression of CsNF-YC6 Enhances Stress Tolerance to Abiotic Factors

To investigate whether *CsNF-YC6* plays a role in drought tolerance, the germination of transgenic Arabidopsis seeds under salt, GA and ABA treatments, the root growth and growth after stress at the adult seedling stage, and physiological index measurements were investigated. The seed germination rate of transgenic line #1 was significantly higher than that of wild type in ABA-stressed medium (Appendix A). The primary root growth of lines #1 and #3 was significantly higher than wild type under ABA stress and GA stress, respectively (Figure 3E,F). After treatment with 200 mM NaCl and 15% PEG, the phenotypes of the wild-type and overexpression lines were significantly different, with the wild-type line showing yellowing and severe chlorotic curling of leaves, while the transgenic lines #1 and #2 showed partial chlorosis and yellowing of leaves, while the transgenic line #3 showed only a small amount of chlorotic curling of leaf edges (Appendix A).

It has been shown that *NF-Y* is involved in regulating the ABA-mediated drought response pathway, and the expression of genes such as *RD29A* is induced by salt and drought stresses. To investigate whether *CsNF-YC6* is involved in the ABA signaling pathway that regulates stress-related genes to enhance plant stress resistance, we selected some genes related to ABA metabolism and stress (*DREB2A*, *NCED3*, *RD29A*, *ABI1* and *ABI5*) and analyzed their expression under salt and PEG drought stress. The stress-related genes were all induced by salt stress and PEG drought stress. Meanwhile, under normal conditions, the transcript levels of *DREB2A*, *NCED3*, *RD29A* and *ABI5* of some transgenic lines were increased to varying degrees (Figure 3G). Under PEG drought simulation conditions, *NCED3* was up-regulated 1.18-fold in transgenic No.1, and *RD29A*, *ABI1* and *ABI5* were up-regulated 3.11, 5.09, 1.11, 1.31, 2.09 and 1.06-fold, in transgenic No.1 and 2, respectively. Under NaCl stress simulation conditions, except *RD29A*, which was up-regulated 1.41, 3.60 and 1.20-fold in the three transgenic plants, the relative expression levels of other abiotic stress-related genes *DREB2A*, *NCED3*, *ABI1* and ABI5 were reduced to varying degrees (Figure 3H–L). The above results can confirm the current speculation that *CsNF-YC6* may be involved in the ABA stress signal transduction pathway, regulating the expression of related genes and thus enhancing the stress resistance of the plants.

The degree of damage to plant cells under abiotic stresses can be reflected by the relative conductivity and malondialdehyde (MDA) content; the more severely damaged the cells, the greater the value. Under normal growth conditions, there was no significant difference in relative conductivity and MDA between the transgenic line and the WT line. After stress treatment, the relative conductivity and malondialdehyde content of the overexpressed and WT lines were higher than those of the plants under normal conditions (Appendix A). The relative conductivity of the WT line was elevated and significantly higher than that of the transgenic line under salt treatment and PEG-mimicked drought conditions. The proline content positively reflected plant stress tolerance. The proline content was significantly increased in all transgenic lines under the stress treatment (Appendix A).

## 3. Discussion

### 3.1. Evolutionary Analysis and Molecular Characterization of the CsNF-YC Gene

Nuclear factors are a family of transcription factors that are ubiquitous in almost all eukaryotes. In yeast and mammals such as humans, each NF-Y branch is encoded by only a single gene member; meanwhile, in plants, the members of different subunit families have been expanded, with different numbers of members in each NF-Y subfamily, with more complex regulatory mechanisms [25]. In general, the number of NF-Y family members shows a positive correlation with the size of the genome. So far, the identification of NF-YC subfamily members has been performed in a variety of plants, for example, in grapevine [25], potatoes [26], rice [5] and poplar [27], and 27, 37, 34 and 42 NF-Y members, respectively, have been identified. In this study, we identified a total of nine tea-plant NF-YC genes, and although the tea plant genome is larger than the genome size of the Arabidopsis, rice and soybean species, the number of NF-YC members identified in tea plants was lower than that of the Arabidopsis, rice, maize and soybean NF-YC subfamily members (numbers of 13, 12, 18 and 15, respectively), but higher than that of oilseed rape and prunus seed (both with 5). Some studies have shown that 10 CsNF-YCs were identified in the genome of the ‘Shuchazao’ tea variety [28]. Through sequence comparison with blast, it was found that CsNF-YC1 and CsNF-YC9 in this study were all corresponding to the CsNF-YC3 that we identified, while other CsNF-YCs were all one-to-one corresponding. The most likely reason is that the genome assembly quality of different tea tree varieties is different, and it may also exist because of the high diversity of tea trees, where some relatively large structural variations may cause the copy number to be different. These suggest that the differences in gene numbers may be related to genetic redundancy and functional differentiation among different NF-YC subfamily members, in addition to variations in genome size or differences in the adaptation mechanisms of different species to the environment. Eight CsNF-Y protein sequences from six tea plant varieties (Longjing 43, Huangdan, Tieguanyin, Biyun, Yunkang 10 and EW) were analyzed using DNAMAN6.0 software online site (Appendix A), and their concordance was 86.67%~97.52%. Among them, CsNF-YC1 had the highest identity and was in complete agreement with the sequence of GWHPASIV009131 of ‘Tieguanyin’, with ‘Huangdan’ and ‘Yunkang 10’ The homologous sequences of CsNF-YC2 had the lowest identity, and the sequence differences with ‘Huangdan’ and ‘Yunkang 10’ were greater, with fragment insertions and three-base or multi-base substitutions. Overall, the sequence differences between the starting and ending sequences of CsNF-YCs were greater, probably due to the quality of genomic sequence splicing; the sequences of CsNF-YCs of ‘Biyun’ genome were more different compared with those of other genomes.

Protein multiple sequence comparison showed that the central structural domain in NF-YC contains about 80 amino acids, which are important for DNA binding activity and can specifically bind to the highly conserved CCAAT sequence in the promoter. In contrast, NF-Y functions mainly in the nucleus by directly binding to CCAAT cis-acting elements in the promoter region of target genes or by interacting with other factors to activate or repress the expression of target genes [29]. Therefore, this conserved structural domain plays an important role in regulating gene expression. Except for *CsNF-YC6*, the number of introns of *CsNF-YCs* is generally low, all containing only one–two introns clustered in one piece in the phylogenetic tree, which is similar to the findings of others [28,30].

Currently, the function of NF-YCs in the model plant Arabidopsis has been more thoroughly studied, and AtNF-YC proteins are extensively involved in plant growth and development such as seedling growth and flowering [19]. Similarly, in our study, many growth- and development-related elements were detected in the upstream promoter of the *CsNF-YC* gene. It is also known that high sequence identity between species can be used to infer similar functions. To predict the potential functions of tea-plant *CsNF-YC* genes, a phylogenetic tree was constructed by combining tea plant *CsNF-YCs* with sequences of NF-YCs of other species with known functions, and we could infer the potential functions of *CsNF-YC* genes homologous to *AtNF-YC* genes. The shape of the obtained tree was similar to that reported in previous studies [28].

### 3.2. CsNF-YC Genes Play an Important Role in Tea Plant Growth and Development and Response to Abiotic Stresses

*NF-Y* regulates plant growth and development, such as controlling hypocotyl growth and endosperm development in Arabidopsis under photoperiod. Studies have shown that plant *NF-Y* genes have tissue expression specificity, such as *ZmNF-YA01*, *ZmNF-YA08* and *ZmNF-YA14* in maize [31] and *RcNF-YB2* and *RcNF-YB12* in castor [32], which show high endosperm and seed development stages in expression levels, involved in regulating embryo formation and seed development. The *CsNF-YC* gene of tea plants is similarly tissue-specific.

Tea plants are subject to endogenous signals or abiotic stresses in their natural growth environment, resulting in reduced growth, yield and quality. Currently, ABA is considered to be the “switch” responsible for initiating shoot dormancy and plays an important role in the seed and shoot dormancy process. This study showed that *DREB2A*, a gene independent of the ABA signaling pathway, was significantly induced under stress and could enhance the drought resistance of plants. However, the expression of *DREB2A* in transgenic lines in this study was significantly down-regulated, which was probably related to the sampling time. In other studies, it was found that the expression level of *DREB2A* gene was significantly up-regulated in the early stage of drought stress, and significantly down-regulated in the late stage of drought stress [33]. Meanwhile, *NF-Y* has been shown to be dependent on the ABA pathway to enhance drought tolerance in plants, but also through non-ABA-dependent pathways such as increased photosynthetic efficiency, antioxidant enzyme activity and reduced hydrogen peroxide content in plants [29]. Under short-term stress, overexpression of *StNF-YC9* increased root length and photosynthetic rate and reduced water loss rate in potato, while under long-term drought stress, *StNF-YC9* reduced malondialdehyde accumulation and enhanced drought tolerance in potato [6]. In this experiment, most *CsNF-YCs* were up-regulated in expression under ABA treatment. In the ABA signaling pathway, *NCED3* is a key gene for ABA synthesis. Poplar *PdNF-YB21* significantly enhanced the expression of *PdNCED3*, which in turn increased ABA content to promote root growth and drought resistance [11]. At the same time, our study found that the expression of some ABA-related genes was down-regulated in transgenic Arabidopsis, which may be because the main mechanism of *NF-YC6* enhancing plant stress resistance is not mainly dependent on ABA signaling, and there may be other pathways.

During the dormancy process in plants, GA acts as a dormancy-breaking agent that can replace the cold-requiring conditions and promote early dormancy breaking. *NF-Y* genes in many plants also respond to GA induction [34]; it has been shown that *NF-YC* can synergistically restrict GA-mediated seed germination with *RGL2* [35]. In addition, studies have shown that *NF-Y* can be used for photoperiodic *CO* and *DELLA* interactions in gibberellin pathways, directly binding to the cis-element of *SOC1*’s promoter to regulate the level of *H3K27* trimethylation to promote flowering [36]. GA induces the expression of most *CsNF-YC* genes, and it is hypothesized that *CsNF-YCs* regulate the growth and development of tea plants through the GA pathway.

### 3.3. Early Flowering and Improved Stress Tolerance by Overexpression of CsNF-YC6

In this experiment, the exogenous hormones ABA and GA were able to induce the expression of the *CsNF-YC6* gene in EW leaves of tea plants at 1.84 and 2.57, respectively, after 36 h of induction, and the apple *MdNF-YC* gene also had a similar expression pattern, with a decrease in expression in the early stages of ABA induction (0–6 h) and a significant increase in the later stage, playing an important role in the biotic stress response in apple [37], suggesting that the *CsNF-YC6* gene may also be involved in regulating abiotic stress mechanisms in tea plants.

*NF-YC* transcription factors can be involved in drought regulation in plants by increasing photosynthetic efficiency, regulating osmotic stress-related physiological processes such as proline accumulation and ROS scavenging, transcript levels of stress genes, and participating in the signaling pathways of different hormones such as ABA [12,38]. Most *NF-Y* transcription factors enhance plant stress tolerance mainly by regulating the transcription levels of genes in stress or ABA signaling pathways; thus, NF-YC can form a heterotrimer with NF-YA and NF-YB to activate the *PYR1*-mediated abscisic acid (ABA) signaling pathway to regulate plant stress tolerance [10]; soybean *GmNF-YA5* overexpression reduces stomatal opening, leaf water loss and increases ABA-dependent pathway genes (*ABI2*, *NCED3*, *LEA3*, *RD29A*) and ABA non-dependent genes (*DREB2A*, *DREB1A*, *DREB2B*) transcript levels [39] and *CdtNF-YC1* plays an important role in drought and stress tolerance in bermudagrass [23].

*ABI5* is an ABA signaling control factor involved in the drought-stress defense response. The expression level of *ABI5* was elevated in *CsNF-YC6* transgenic plants under PEG-simulated drought conditions, and in addition, *ABI1*, a transcriptional repressor in the ABA signaling pathway, was found to be significantly down-regulated under PEG-simulated drought stress and NaCl stress. The *NF-YC* of the remaining plants also had similar functions [23]. Meanwhile, the gene *DREB2A*, which is not dependent on the ABA signaling pathway, was significantly induced. In conclusion, overexpression of the *CsNF-YC6* gene enhances the expression of stress-responsive genes, activates the transcriptional levels of ABA signaling genes, and ultimately improves the tolerance of transgenic Arabidopsis to salt and drought stresses.

Numerous studies have shown that *NF-Y* transcription factors are involved in plant growth and development. *CsNF-YCs* may be involved in tea plant development by regulating target genes in multiple physiological pathways, including photosynthesis, chlorophyll metabolism, fatty acid biosynthesis, and amino acid metabolism pathways [28]. In this experiment, it was found that the number of cauline leaves and silique of overexpression lines during normal growth was greater than wild type, the stem length was also significantly higher than wild type, the growth and development were better than wild type and overexpression of *CsNF-YC6* could promote root and above-ground growth of Arabidopsis seedlings. Flowering is an important event in plant growth and development, and in the model plant Arabidopsis *AtNF-Y* physically binds to the CCAAT element in the FT promoter and forms a CO-NFY complex with CO to bind other proteins, which then synergistically interacts with *TCP7* to activate the flowering integrator gene *SOC1* to accelerate flowering [40]. In this experiment, overexpression plants also flowered earlier than the wild type. It is hypothesized that *CsNF-YC6* can be involved in the growth and development of Arabidopsis.

## 4. Materials and Methods

### 4.1. Plant Materials, Growth Conditions, and Treatments

The different tissues and tea plant branches and other materials were all 4–6 years old clonal ‘Emei Wenchun’ (EW) plants, which were harvested from Meishan, Sichuan Province. Overwintering axillary buds and leaf materials of EW and ‘Chuancha 2’ (CC2) were harvested from Muchuan, Sichuan Province, and the sampling time was concentrated during the overwintering period of tea plants (November 2021–February 2022) at 15-d intervals.

Branches were precultured in 1/2 MS liquid medium. The culture conditions were constant temperature (23 ± 0.5 °C), 80% humidity, light (12 h)/dark (12 h) and light intensity of 8000 lx in an artificial culture room. Then, 7 d after preculture, plants with similar growth status were selected for hormone treatment. The hormone treatments were foliar sprays of 50 mg/L gibberellic acid (GA) and 100 μM abscisic acid (ABA) in aqueous solution, with 5–6 tea seedlings per group treatment. Mature leaves at the 2nd–3rd leaf position were taken as test material at 6 time points of treatment 0, 6, 12, 24, 36 and 72 h. All samples were snap frozen in liquid nitrogen immediately after collection and stored at −80 °C. Each treatment contained three biological replicates.

The salt stress was applied to *Arabidopsis thaliana* in 1/2 MS medium (pH = 5.7) containing 200 mM NaCl, and the PEG treatment was applied to *Arabidopsis thaliana* in 1/2 MS nutrient solution containing 15% PEG. Samples were taken after 7 d of both stress treatments.

### 4.2. Identification, Phylogenetic Tree Construction, Gene Structure, Motif Analysis and Multiple Comparisons

The tea plant ‘Longjing 43’ genome sequence, CDS sequence, protein sequence and its annotation information were obtained from the National Center for Biological Information website (https://www.ncbi.nlm.nih.gov/, accessed on 10 September 2021); the model plant Arabidopsis AtNF-YC-related protein sequences were obtained from the Tair database (http://www.arabidopsis.org/, accessed on 10 September 2021). The local tea-plant gene database was constructed using Blast (v2.13.0) software downloaded from NCBI, and the protein sequences of Arabidopsis NF-YC family members were used as queries, blastp to search the local database and the E value was set to 0.001. According to the Pfam number (PF00808) downloaded from the Pfam database (https://pfam.xfam.org/, accessed on 10 September 2021), the local database was retrieved using HMMER software (V3.2.1), and the E value was set to 0.01.

After screening out protein sequences without intact coding regions, SMART (http://smart.embl-heidelberg.de, accessed on 10 September 2021) and CDD (http://www.ncbi.nlm.nih.gov/cdd, accessed on 10 September 2021) websites were used for subsequent structural domain analysis to remove sequences without intact structural domains, and sequences containing intact NF-YC domains were identified as CsNF-YCs, and the sequences were named according to their position distribution on the chromosome. The amino acid number, isoelectric point and molecular weight were predicted using Expasy (https://www.expasy.org/, accessed on 10 September 2021). The number and position of protein structural domains were predicted using the CDD tool in NCBI, and MEME 5.1.1 website was used for analysis of conserved motifs and TBtools for mapping. The gene structure of *CsNF-YCs* was visualized using the online tools GSDS 2.0 and TBtools (v2.028). The 2000 bp sequences upstream of *CsNF-YCs* were obtained and submitted to PlantCARE to analyze the cis-acting elements in their promoter regions. The *CsNF-YC* genes sequences were submitted to MEGA 6.06 to construct a gene family phylogenetic tree; the downloaded Arabidopsis, oilseed rape and Tribulus terrestris protein sequences were compared with those of tea plants using Clustal W. The *NF-YC* sequences of Arabidopsis, oilseed rape, and Tribulus alfalfa were downloaded from the PlantTFDB online website (http://planttfdb.gao-lab.org, accessed on 10 September 2021). Phylogeny was selected for neighbor-joining (NJ) tree construction, and Bootstrap was selected as 1000.

### 4.3. Quantitative Real-Time PCR

EASY spin PLUS RNA Kit (Aidlab, Beijing, China) was used to extract total RNA from tea-plant samples of different tissues and treatments; cDNA was synthesized according to FastKing cDNA First Strand Synthesis Reagent (Tiangen, Beijing, China). qRT-PCR reactions were performed in Bio-Rad CFX96 (Bio-Rad, Hercules, CA, USA). Specific systems and procedures were carried out according to the instructions of Universal Blue qPCR SYBR Green Master Mix (YEASEN, Shanghai, China). The relative expression levels of the genes were calculated by the 2^−ΔΔCt^ method using tea *CsGAPDH* as the internal reference gene. The primers are illustrated in Appendix A.

### 4.4. Cloning and Physicochemical Property Analysis of CsNF-YC Gene

The coding sequences of *CsNF-YCs* were cloned from cDNAs of mixed tissue samples of EW. The data of five publicly available tea plant genomes (Longjing 43, Huangdan, Tieguanyin, Biyun and Yunkang 10) were downloaded and homologous genes of EW-cloned *CsNF-YCs* were searched for using local Blast and analyzed for comparative identity. Open reading frames and protein sequences were predicted by DNAMAN6.0 software. Protparam (http://web.expasy.org/protparam/, accessed on 5 March 2022) predicted the molecular weight and theoretical isoelectric point of CsNF-YC proteins. CBS (http://www.cbs.dtu.dk/services/TMHMM-2.0/, accessed on 12 March 2022) as well as SignalP (http://www.cbs.dtu.dk/services/SignalP/, accessed on 28 March 2022) were used for NF-YC proteins for transmembrane and signal peptide analysis. Wolf PSORT (http://wolfpsort.hgc.jp/, accessed on 6 April 2022) was used for the subcellular localization prediction of CsNF-YC proteins. CD-search and MEME search were used for conserved motifs. Psipred (http://bioinf.cs.ucl.ac.uk/psipred/, accessed on 10 April 2022) and SWISS-MODEL (http://swissmodel.expasy.org/, accessed on 20 April 2022) predicted the secondary and tertiary structures of CsNF-YC protein junctions.

### 4.5. Expression Vector Construction and Agrobacterium-Mediated Transformation of Arabidopsis thaliana

The coding sequence of *CsNF-YC6* was inserted into the expression vector pCAMBIA1300-35S-eGFP between SacI and SalI restriction sites by homologous recombination method. ClonExpress II One Step Cloning Kit (Vazyme, Nanjing, China) is used for homologous recombination. Based on the concentration of the recovered product, the volume required for the carrier and the inserted fragment is calculated. The recombinant product was transformed into *E. coli* DH5α and then into *Agrobacterium tumefaciens* GV3101. Young tobacco was injected using the leaf dorsal injection method, incubated in the dark for 48 h in the incubator and subcellular localization was observed by fluorescence microscopy after 20 min under normal light. An overexpression vector (*35S:CsNF-YC6-pCAMBIA1302*) was constructed and transformed into Agrobacterium strain EHA105, and transgenic Arabidopsis plants were obtained by inflorescence infestation method and screened using thaumatin B (*Hyg*). Lines with healthy cotyledon on the medium were transplanted into pots, and DNA was extracted after 30 days of growth, and hygromycin gene fragments were amplified with gene-specific primers. In addition, the expression level of *CsNF-YC6* in transgenic Arabidopsis plants was detected using qRT-PCR.

### 4.6. Abiotic Stresses Treatment

A 1/2 MS stress medium with 150 mM NaCl, 0.5 μM ABA and 0.5 μM GA was prepared. The seeds were sterilized, vernalized and spotted on the medium with different stress treatments (GA, ABA, NaCl), and the seed germination rate was counted after 4 d of light culture and photos were collected. After being spotted on the 1/2 MS blank medium and cultured vertically with light for 4 d, seedlings of uniform growth were transplanted and cultured vertically on the new blank medium and stress treatment medium, and root length was counted and photos were collected after 10 d. After 4 weeks of planting Arabidopsis, 1/2 MS nutrient solution of salt stress and PEG treatment was watered to Arabidopsis. Samples were taken after 7 d of both stress treatments.

### 4.7. Morphological Characteristics of Transgenic Arabidopsis Plants

Wild-type and overexpression lines were cultured in nutrient soil for 4 weeks and then morphologically characterized. The stem length of the Arabidopsis lines was measured with a straight edge.

### 4.8. Measurement of Physiological Indicators

The relative conductivity was determined by weighing 0.1 g of cut leaves in a 50 mL centrifuge tube, adding 10 mL of ultrapure water and measuring the conductivity E0 after 12 h. The conductivity E1 was measured after heating in a boiling water bath for half an hour and the conductivity of the distilled water was ECK as the control. The method for determination of proline content: Weigh 0.5 g Arabidopsis leaves, place them in 10 mL centrifuge tube, add 5 mL 3% sulfosalicylic acid, extract them in boiling water for 10 min, and then filter them after cooling. After cooling, add 4 mL of toluene and shake for 30 s. After standing, centrifuge the upper layer at 3000 r/min for 5 min. The homogenate was mixed with 8.5 mL of 10% TCA and centrifuged at 3000× *g* for 10 min. Using water as the blank, 3 mL of supernatant and 3 mL of water were taken separately, and 3 mL of 0.5% TBA solution was added and shaken well. The solution was boiled for 10 min and then cooled immediately (the small bubbles of the solution were timed). The absorbance values of the samples were measured at 430 nm, 532 nm and 600 nm with the blank as the reference.

### 4.9. Experimental Data Processing and Statistical Analysis

Data were analyzed using the one-way analysis of variance (ANOVA) method with SPSS 22 software, and bar and line graphs were drawn using GraphPad Prism 8.

## 5. Conclusions

In this study, nine genes were identified in the reference genome of tea plant ‘Longjing 43’, and comprehensive structural characterizations, phylogenetic analyses and expression analyses were performed. In general, there was little difference in homologous gene sequences among the different tea cultivars. Due to the influence of genome assembly quality, it is necessary to further clone the *CsNF-YC* gene sequences of different tea varieties and then compare whether there are sequence differences or different cuts of each gene in different tea plant varieties. In tea plants, exogenous ABA induces the expression of most *CsNF-YC* genes, and we hypothesized that *CsNF-YC* genes are involved in ABA-mediated stress. GA can induce the expression of most *CsNF-YC* genes, suggesting that *CsNF-YC* regulates the growth and development of tea plants through the GA pathway. Under ABA and GA treatments, the seed germination rate and root length of *CsNF-YC6* transgenic plants increased. In addition, this resulted in an increased flowering period in Arabidopsis. Flowering pathway-related genes were up-regulated in the *CsNF-YC6* overexpression lines, which in turn were involved in the physiological process of flowering in Arabidopsis. These results provide a basis for understanding the evolution of *CsNF-YC* genes and their potential roles in the growth and development of tea plants and abiotic stresses. 

## Figures and Tables

**Figure 1 ijms-25-00836-f001:**
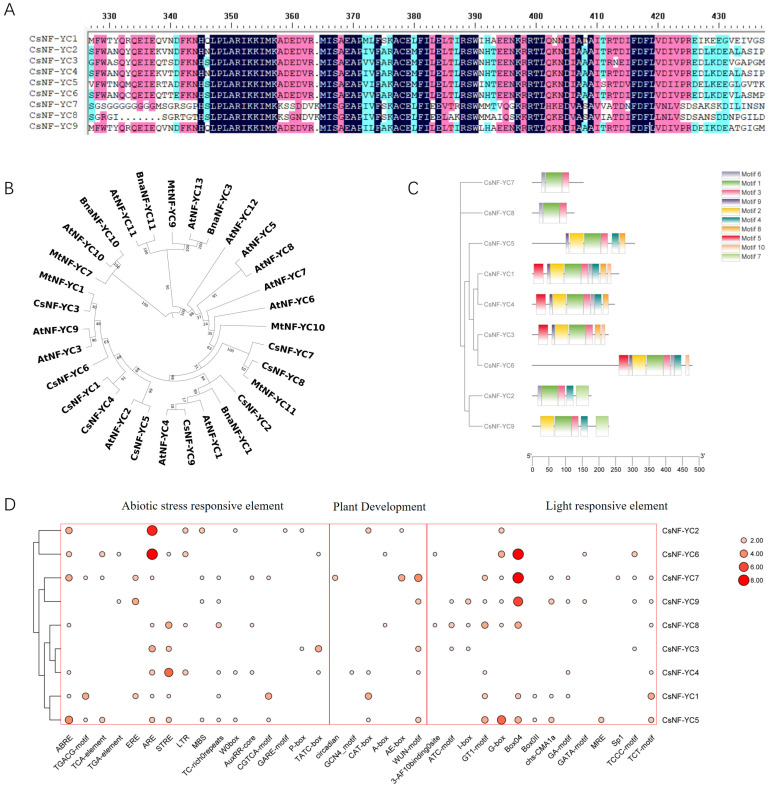
Bioinformatics analysis of CSNF-YCs. (**A**) Multiple sequence alignment of different CsNF-YC subfamily proteins; (**B**) phylogenetic analysis of NF-YCs subunit family in Tea plant, Arabidopsis, Brassica napus and Medicago truncatula. Cs represents *Camellia sinensis* L.; At represents *Arabidopsis thaliana;* Bn represents *Brassica napus* L. and Mt represents *Medicago truncatula*; (**C**) motif analysis of *CsNF-YC* genes; (**D**) distribution of cis-acting elements associated in the *CsNF-YCs*.

**Figure 2 ijms-25-00836-f002:**
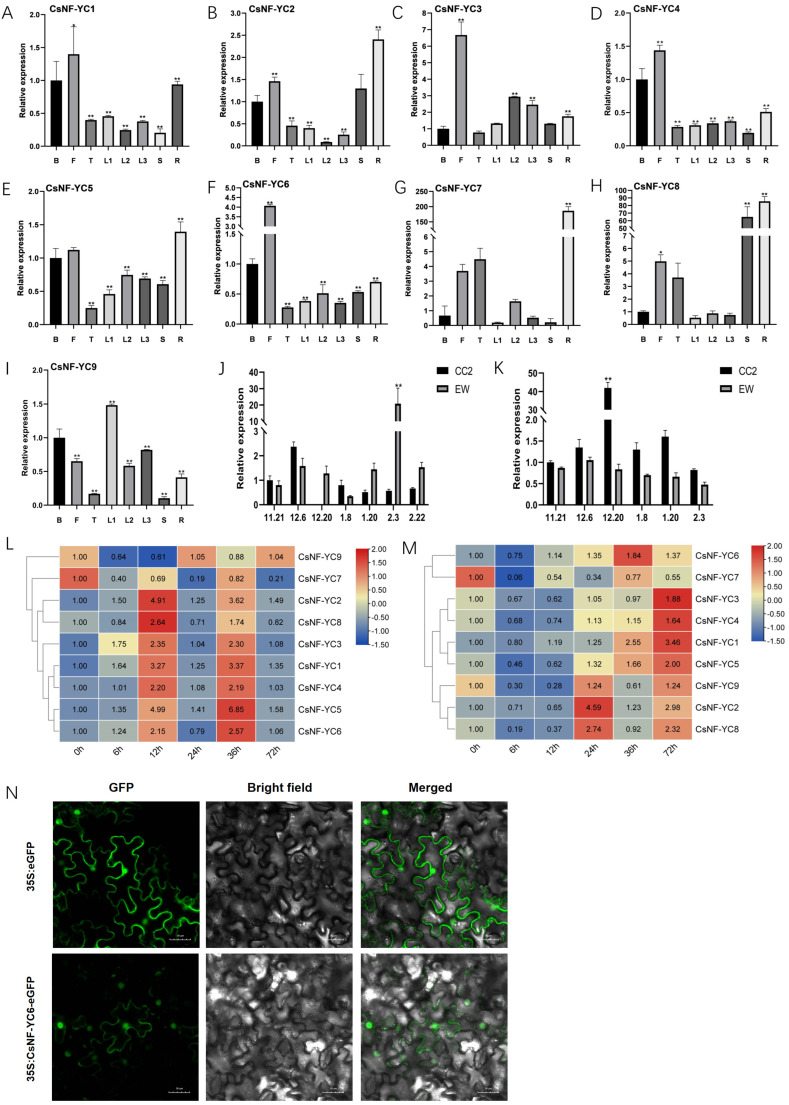
Expression pattern and subcellular localization of *CsNFYC* gene family. (**A**–**I**) Tissue-specific expression pattens of *CsNF-YC* genes. The surveyed tissues include bud (**B**), flower (**F**), terminal bud (T), younger leaf (L1), old leaf (L2), mature leaf (L3), stem (S) and root (R). The transcript levels of each gene in various tissues were compared with bud. (* *p* < 0.05, ** *p* < 0.01); (**J**,**K**) Dynamic changes of expression of *CsNF-YC6* during over-wintering. Leaves (**J**); axillary bud (**K**); (**L**,**M**) expression analysis of *CsNF-YC* genes under GA (**L**) and ABA (**M**) treatments in leaves. The samples that were treated with 0 h were set as control, The value on the heat map is 2^−ΔΔCt^; (**N**) subcellular localization analysis of *CsNF-YC6*.

**Figure 3 ijms-25-00836-f003:**
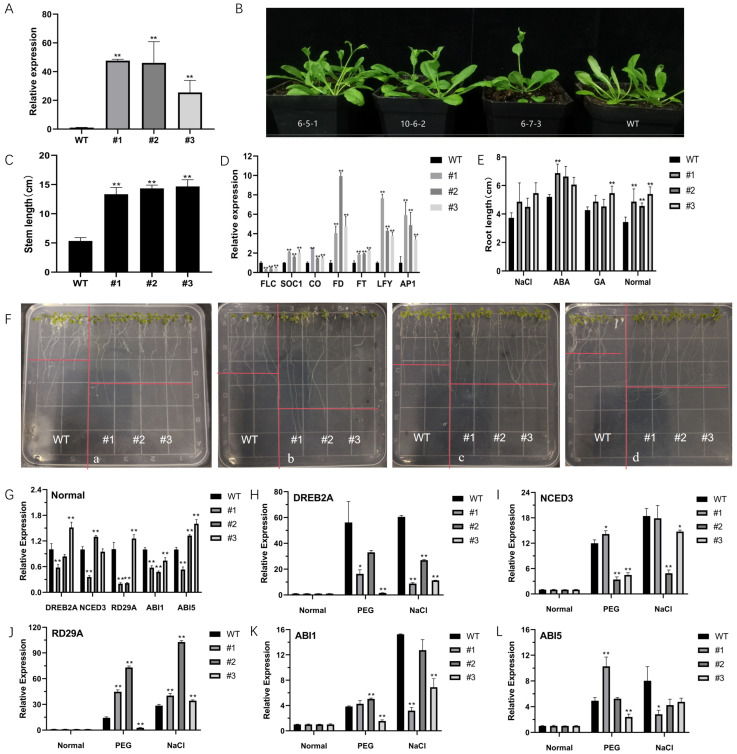
Phenotypic analysis and expression patterns of *CsNF-YC6* transgenic Arabidopsis plants. (**A**) Expression levels of *CsNF-YC6* in Arabidopsis; (**B**) phenotype of *Arabidopsis thaliana* grown for 30 d in long daylight; (**C**) stem lengths were calculated in Arabidopsis; (**D**) expression levels of flowering-related genes in Arabidopsis; (**E**) root elongation of *CsNF-YC6-OE* Arabidopsis lines under different stress conditions; (**F**) the phenotype of WT and three lines grown in 150 uM NaCl (**a**), 0.5 uM ABA (**b**), 0.5 uM GA (**c**) and 1/2 MS (**d**) medium for 10 days; (**G**) stress gene analysis of WT and three lines under normal culture conditions. Normal WT was used as control. (* *p* < 0.05, ** *p* < 0.01). (**H**–**L**) stress gene relative expression analysis of WT and three lines under 200 mM NaCl treatment and 15% PEG treatment. DREB2A (**H**), NCED3 (**I**), RD29A (**J**), ABI1 (**K**) and ABI5 (**L**). Normal WT and 3 strains were used as control (the relative expression level normalized to 1), and the wild type after stress treatment was used as comparison object. (* *p* < 0.05, ** *p* < 0.01).

## Data Availability

All the data presented in this study are included in the manuscript and Appendix A.

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
