# Peer review of "Genome-Wide Analysis of Nuclear factor-YC Genes in the Tea Plant (Camellia sinensis) and Functional Identification of CsNF-YC6"

_ijms, 2024, doi:10.3390/ijms25020836_

Round 1
Reviewer 1 Report
Comments and Suggestions for Authors
The paper is interesting and provides novel insights into a family of transcription factors, but some aspects need major improvement before publications could be recommended.
The striking point is that stress response genes seem to be down regulated in transgenic plants. Could it be that the main mechanism of these transcription factor is not dependent on the ABA signalling and that plants are sensing lees stress? Please, comment this possibility. Another problem may be that stress is not well applied. Please include a novel recalculating the data in figure 3H, but comparing the expression of each gene with the same gene under NaCl or PEG, and normalizing to 1 the control conditions. In this case it could be observed whether there is a real induction upon stress and evaluate better what is going on in the transgenic plants.
Binnomial names should be given in italics.
Figure 3: enlarge panels F and G, or make a different figure with them, as it is difficult to see anyting and the mentioned phenotype is not really visible.
Comments on the Quality of English Language
english needs minor stilistic correction
Author Response
请参阅附件。

Reviewer 2 Report
Comments and Suggestions for Authors
This study offers valuable insights into the CsNF-YC gene family in tea plants, particularly highlighting the multifaceted role of CsNF-YC6. By elucidating its impact on growth, development, and stress responses, this research expands our understanding of plant biology and underscores CsNF-YC6's significance in orchestrating key physiological processes. While this manuscript presents valuable insights, further revisions are necessary before considering it for acceptance. Addressing specific points related to clarity, methodology, and data interpretation will bolster its suitability for publication.
Comments:
1. In abstract, “and chromosomal localization of this gene were analyzed.” In the previous sentence, nine genes were referenced. To which specific gene does the term "this gene" refer? Please revise it for improved clarity.
2. Abbreviations should be defined at first mention. Please revise “GA and ABA” in abstract; on page 2, “HFM structural domain”, “H2A”, “FT promoter”, “GRAS proteins”; on page 4, “POTS”.
3. Several sentences could benefit from revision to enhance the overall flow and coherence of the text. In abstract, “This study knowledge of CsNF-YCs genes and reveals that CsNF-YC6 plays important roles in plant growth, root and flower development and response to abiotic stresses.”; on page 2, “In plants, NF-Ys can regulate plant growth and development by participating in a variety of physiological and developmental processes, involved in the entire plant growth and development process from nutritional to reproductive growth including seed germi- nation, root growth, and plant flowering”; on page 3, “Universal Blue qPCR SYBR Green Master Mix (YEASEN, Shanghai, China) parameters were 95°C pre-denaturation for 5 min; 95°C denaturation for 10 s, 58°C annealing for 20 s, 72°C extension for 20 s, 40- 45 cycles”; on page 5,
“the proline content was determined by weighing 0.5 g of each leaf of Arabidopsis thaliana in a 10 mL centri- fuge tube, adding 3% sulfosalicylic acid 5 mL the leaves were extracted with boiling water for 10 min, cooled and filtered.”
4. Some spaces are missing on page 1“et al., 2021a);10,”; “07);6 NF-YA”.
5. Some terms need to be italic. “Arabidopsis thaliana” on page 3, 5, 11; “Agrobacterium tumefaciens” on page 4.
6. on page 4, “The coding sequence of CsNF-YC6 was inserted into the expression vector pCAM- BIA1300-35S-eGFP between SacI and SalI restriction sites by homologous recombination method.” Homologous recombination differs from typical molecular cloning methods. Could the authors provide more detail, such as the enzyme used in this process for clarity?
7. In Arabidopsis, 'line' is the preferred term over 'strain'. Please change all strain to line on page 4, 10, 11.
8. In section 2.6, what concentration of GA is being utilized in this context?
9. In section 3.1, it's stated that 'a total of nine CsNF-YC family members were identified, named CsNF-YC1 to CsNF-YC9.' However, a prior study by Wang et al. in 2019 identified 10 CsNF-YCs in their research titled 'Identification, expression, and putative target gene analysis of nuclear factor-Y (NF-Y) transcription factors in tea plant (Camellia sinensis).' It would be beneficial to compare and discuss these findings in the discussion section for comprehensive analysis.
10. On page 6, it's mentioned that 'excluding CsNF-YC2, CsNF-YC7, and CsNF-YC8, all other members contain Motif 2 and Motif 4 motifs.' However, in Figure 1C, it's evident that CsNF-YC2 also contains Motif 4.
11. On page 8, “then decreasing, then decreasing,” “tea tree leaves. in tea leaves”, are these typos?
12. In Figure 2N, Please ensure that the photos a and b are displayed at the same magnification and include the bright fields for consistency.
13. On page 11, “After treatment with 200 mM NaCl and 15% PEG, the phenotypes of the wild-type and overexpression strains were significantly different, with the wild-type strain showing yellowing and severe chlorotic curling of leaves, while the transgenic strains #1 and #2 showed partial chlorosis and yellowing of leaves, while the transgenic strain #3 showed only a small amount of chlorotic curling of leaf edges.” Can you include the data/photo to support this?
14. Please include the line number in the revision.
Comments on the Quality of English LanguageModerate editing of English language required. Some sentences need to be revised for improved clarity. Several sentences could benefit from revisions to enhance clarity.
Round 2
Reviewer 1 Report
Comments and Suggestions for Authors
Authors have considered my suggestions and the paper has substatntially improved.
I can recommend publication
Reviewer 2 Report
Comments and Suggestions for Authors
Most comments have been addressed, significantly enhancing the manuscript except for comment 12. I am looking forward to see the revision for comment 12.
There are a few minor changes remaining:
1. In line 5, please remove "." after 'flowering.'.
2. In line 198, there appears to be missing spaces in 'add 5mL3%de sulfosalicylic acid.' Additionally, could it be a typo for 'de sulfosalicylic acid'?"
Round 3
Reviewer 2 Report
Comments and Suggestions for Authors
Thank you for the revised submission. Overall, the revision seems to meet the required standards.